# The Evolving Landscape of the Molecular Epidemiology of Malignant Pleural Mesothelioma

**DOI:** 10.3390/jcm10051034

**Published:** 2021-03-03

**Authors:** Sara Lettieri, Chandra Bortolotto, Francesco Agustoni, Filippo Lococo, Andrea Lancia, Patrizia Comoli, Angelo G. Corsico, Giulia M. Stella

**Affiliations:** 1Department of Medical Sciences and Infective Diseases, Unit of Respiratory Diseases, IRCCS Policlinico San Matteo Foundation, University of Pavia Medical School, 27100 Pavia, Italy; sara.lettieri01@universitadipavia.it (S.L.); corsico@unipv.it (A.G.C.); 2Department of Intensive Medicine, Unit of Radiology, IRCCS Policlinico San Matteo Foundation, University of Pavia Medical School, 27100 Pavia, Italy; c.bortolotto@smatteo.pv.it; 3Department of Medical Sciences and Infective Diseases, Unit of Oncology, IRCCS Policlinico San Matteo Foundation, University of Pavia Medical School, 27100 Pavia, Italy; f.agustoni@smatteo.pv.it; 4Thoracic Unit, Catholic University of the Sacred Heart, Fondazione Policinico A. Gemelli, 00100 Rome, Italy; filippo_lococo@yahoo.it; 5Department of Intensive Medicine, Unit of Radiation Therapy, IRCCS Policlinico San Matteo Foundation, University of Pavia Medical School, 27100 Pavia, Italy; a.lancia@smatteo.pv.it; 6Cell Factory and Pediatric Hematology-Oncology Unit, IRCCS Fondazione Policlinico San Matteo, 27100 Pavia, Italy; p.comoli@smatteo.pv.it

**Keywords:** mesothelioma, genetics, molecular epidemiology, personalized medicine

## Abstract

Malignant pleural mesothelioma (MPM) is a rare and aggressive malignancy that most commonly affects the pleural lining of the lungs. It has a strong association with exposure to biopersistent fibers, mainly asbestos (80% of cases) and—in specific geographic regions—erionite, zeolites, ophiolites, and fluoro-edenite. Individuals with a chronic exposure to asbestos generally have a long latency with no or few symptoms. Then, when patients do become symptomatic, they present with advanced disease and a worse overall survival (about 13/15 months). The fibers from industrial production not only pose a substantial risk to workers, but also to their relatives and to the surrounding community. Modern targeted therapies that have shown benefit in other human tumors have thus far failed in MPM. Overall, MPM has been listed as orphan disease by the European Union. However, molecular high-throughput profiling is currently unveiling novel biomarkers and actionable targets. We here discuss the natural evolution, mainly focusing on the novel concept of molecular epidemiology. The application of innovative endpoints, quantification of genetic damages, and definition of genetic susceptibility are reviewed, with the ultimate goal to point out new tools for screening of exposed subject and for designing more efficient diagnostic and therapeutic strategies.

## 1. Introduction

Malignant mesothelioma (MM) is a rare cancer that arises from the cells that line the pleura, the pericardium, and/or peritoneum. Indeed, the cells that define the mesothelium derive from the mesodermal coelomic cavity of the embryo. The tissue develops as a continuous epithelial sheet that lines the pleura, pericardium, and peritoneal cavity. A single layer of mesothelial cells rests on the basement membrane [1]. Their mitotic index is slightly low, but they can grow in response to inflammatory/malignant damage. Malignant pleural mesothelioma (MPM) identifies a primary neoplastic lesion arising from the pleural layer and is more often unilateral at diagnosis. MPM usually affects patients from the fifth to seventh decades, and in 70–80% of cases it develops in males. The tumor can invade both the visceral and parietal pleura and frequently extend to adjacent structures. Prognosis is extremely poor, with a median survival of about one year from diagnosis [2]. It was almost unknown up to the beginning of the 19th century, but its incidence has dramatically increased since then due to large-scale asbestos mining and a more frequent industrial use thanks to its chemical and physical properties. Indeed, the increasing use of this mineral since the Second World War has led to the description of a causal relationship between exposure to asbestos and the development of mesothelioma in the 1960s [2,3]. Although asbestos was banned from the Western world by the 1980s, long latency period between exposure to asbestos fibers and MPM onset as well as the slow and complicated process of asbestos disposal explain why the mortality rate from mesothelioma is continuously increasing [4]. Recently the World Health Organization has estimated around 125 million people who are currently exposed to asbestos in their workplace and around 90,000 deaths each year from mesothelioma worldwide (website at https://www.who.int/news-room/fact-sheets/detail/asbestos-elimination-of-asbestos-related-diseases (accessed on 1 December 2020))Although asbestos is the major causative agent of MPM onset, the oncogenic potential of other biopersistent fibers has been reported in some geographic areas as well as in some work settings. Simian Virus 40 (SV40) infection was previously explored as aetiologic agent, but its role was not definitively proven [5]. Moreover, in about 20% of cases, asbestos exposure is not referred or documentable. In parallel to the emergence of novel MPM epidemic clusters, in the last years, research progress has led to increased knowledge of the biomolecular disease features. The integration of molecular techniques into epidemiologic studies may provide new insights and has been referred to as molecular epidemiology [6]. Overall, molecular epidemiology strategies applied to cancer aim at improving patient outcome and at designing effective screening programs. This approach is of extreme relevance for an orphan disease as MPM, since it allows the detection of gene mutations associated to high cancer risk as well as the emergence of potentially actionable targets.

## 2. Why to Apply Molecular Epidemiology to MPM

Since the completion of the Human Genome project in 2003, a major challenge has been the molecular characterization of human cancers. *Omics*-based analysis has now become exploitable and has enabled a rapid identification of potential actionable markers. The genetic characterization of MPM has not been fully clarified until now due to the significant inter-patient variability and to the scarcely reported genetic aberrations [3]. Indeed, although relevant advances have been made regarding genetic drivers of epithelial malignant transformation, very few results are available on pleural cells, which originates from embryo mesoderm-derived cell transformation. Overall, for these reasons, MPM is still recorded as orphan disease by the European Union (EU, website http://www.rarecarenet.eu/ (accessed on 1 December 2020). Therefore, at present, no biotargets are effectively used against MPM. The need for new treatments is therefore urgent and great effort is mandatory to deeper characterize the molecular basis and the genetic tracts that define the disease. Growing evidence suggests that asbestos-induced inflammation is involved in mesothelial malignant transformation and that the crosstalk between neoplastic cells and their unique tumor microenvironment can modulate response to therapies.

In this context, which lacks clear oncogenic drivers, gene expression analysis, rather that detection of single point mutation, emerges as an exploitable strategy to identify those subjects at higher risk of disease progression and worse outcome. The pathogenic players in MPM onset are represented in Figure 1A. In this perspective, molecular epidemiology focuses on the trend of molecular biomarkers in epidemiological settings. This approach—which was initially defined at the beginning of the 1980s [7]—has the goal of clarifying disease aetiology in order to stratify exposed population according to the risk of developing cancer and to identify genetic factors for cancer inter-individual susceptibility [8]. Therefore, molecular epidemiology defines the distribution and determinants of diseases in human populations by descriptive and analytical approaches. Traditional epidemiology is defined as the study of the distribution and determinants of health-related states or events in specified populations, as well as the application of this study to the control of health problems [9]. Its limitation is mainly related to its weak contribution in understanding complex diseases and exposures, since its focus is specifically the public health and general population with a top-down (structural) strategy [10]. Technical progresses and the increasing knowledge on biological mechanisms have paved the way to genetic epidemiology, which specifically studies inherited susceptibility to disease as well as to molecular epidemiology, which is more generally focused on biological markers of risk or exposure, including genes (somatic or germline). In this context, a disease biomarker is defined as a substance, structure or process that can be measured (through an assay) in the human body and may influence the incidence or outcome of that disease [11]. Biomarkers can be classified as of exposure, disease, and susceptibility. The aim of measuring exposure biomarkers is that to decrease exposure misclassification and to determine biologically relevant exposures as well. However, significant limitation is related to the difficulties in timing the pathogenic relevance and in defining exposure rates in the past. Disease markers are represented by somatic gene mutations as well as epigenetic alterations that are associated with disease onset. Disease susceptibility markers might deal with high penetrance genes, with low prevalence in the general population and limited impact on overall cancer burden, or low penetrance genes featuring high prevalence in general population, thus impacting on environmental and endogenous carcinogens [12,13,14]. Overall, the main limitations of biomarker-related studies are the following: (i) some markers might be not very reliable; (ii) their biological meaning is not always clear; (iii) a long gap should be needed between marker development and its validation; (iv) there is a time relationship between exposure, marker measurement, and disease; (v) more frequently, only one bio-sample is available, and little is known about intra and inter-individual variation; and (vi) little is known on potential confounders. The advent of various multi-omics biotechnology platforms and high-throughput techniques has allowed for a series of advantages, among which the possibility to work on quantitative rather than discrete markers, which are more related to phenotype. Therefore, it is possible to measure many markers simultaneously and to define individual signatures that can be used to predict the onset of the disease. Among other -*omics*, of note is the recent development of “radiomics”, which extracts a large amount of information from biomedical images from different sources (CT (computed tomography), MRI (magnetic resonance imaging), PET (positron emission tomography)/CT, US (ultrasound scan)) using data characterization algorithms [15]. This information, named “radiomic features”, can be used for diagnostic purposes, treatment selection, or as prognostic biomarkers. A specific subset of studies focuses on “radiogenomics”, a term that refers both to genetic variation in normal tissues associated with different responses to radiation therapy and to the correlation between cancer imaging and genetic features (Imaging Genomics). Imaging Genomics signatures can be obtained from different imaging modalities and can be integrated into multiparametric models including biological and epidemiological data.

Specific case trials (case-cohort/case-control) should be designed on the basis of marker proprieties (sensibility to batch, storage, and freeze–thaw cycles). The molecular epidemiology approach has a relevant potential in the study of MPM since it can allow for a deeper understanding of susceptibility factors in order to design an effective screening program in occupationally or environmentally exposed populations and improving diagnostic and therapeutic tools. Within respect to an orphan disease such as MPM, novel immunohistochemical and molecular markers have improved the accuracy of diagnosis; however, a high percentage of MPM diagnoses (from 14% in developed countries to 50% in the poorest ones) remain incorrect, impairing epidemiologic approaches [16]. Here, we discuss and analyze how evolving understanding of the dynamic crosstalk between disease exposure (asbestos), susceptibility (germline genetic asset), and disease (somatic events) biomarkers can have an impact on the knowledge of MPM pathogenic mechanisms.

## 3. Asbestos: Above and Beyond

Undoubtedly the most important causative agent for the development of mesothelioma is asbestos, which undeniably represents the most known exposure biomarker. It is completely safe in its natural state as a solid-state rock. It represents a series of naturally occurring silicate minerals, commercially exploited for their properties such as sound absorption and resistance to traction, flame and heat, electrical, and chemical damage. Asbestos fibers can be divided into two main groups: serpentines and amphiboles. Serpentines have only one subtype, chrysotile, also called white asbestos for the light color. These fibers are short and curved and represent about 95% of all asbestos used in industry. Amphiboles include several subtypes, such as crocidolite (blue asbestos), amosite (brown asbestos), and tremolite. They are long and needle-like fibers. Epidemiological data suggest that amphiboles are associated with the highest risk of mesothelioma and that serpentines are associated with lower risk. International Agency for Research on Cancer (IARC) has classified both serpentines and amphiboles as Class I carcinogens [17,18]. Like coal dust in coal mines, asbestos fibers, due to their nanosize, impact on the lymphatic stomata, which are opened on the parietal layer of the pleura, determining the so-called “black spots”. Mesothelial cells have been shown to be 10 times more sensitive to the cytotoxic effects directly caused by the asbestos fibers if compared with bronchial epitelium cells [19]. The persistent interaction of fibers within mesothelial layer is responsible for the development of reactive oxygen species (ROS), which can cause DNA mutations and breakdown of chromosomal chains leading to cell apoptosis and compensatory proliferative stimulus. The result of the above-mentioned processes is malignant transformation [20]. The latency from exposure to the overt disease, and patient death extends over an interval between 14 and 72 years (mean: 48.7 years), with a wide variability related to the intensity of exposure and the type of fiber. The diagnoses of mesothelioma are largely placed in subjects with occupational exposure to asbestos; nevertheless, there is evidence that mesothelioma is linked to both “para-occupational” exposure (e.g., wives who washed clothing from work of husbands who worked around asbestos), and to environmental exposure. Many epidemiologic studies have estimated the population attributable fractions (PAFs [21]) for asbestos and mesothelioma. The PAF is the measure of a fraction of disease in a population attributable to exposure to an environmental agent and depends on the risk associated with the exposure and the fraction of the population exposed to the agent. Previous studies related to the PAF in asbestos-exposed patients who developed mesotheliomas showed that about 88% of pleural mesothelioma and 58% of peritoneal mesotheliomas in men could be related to exposure to asbestos, whereas in women, the results were less conclusive, with only 23% of all mesotheliomas being attributable to asbestos [22]. This work provided strong evidence on one hand for the causative role of asbestos, whereas on the other, it pointed out a gender-related susceptibility. Notably, a fraction of mesothelioma occurs in absence of asbestos exposure. Indeed, although rarely, an idiopathic or spontaneous form of mesothelioma can develop in individuals with no history of exposure to asbestos. Because of the extensive use of asbestos, it is more likely that a fraction of those cases thought to be unrelated with asbestos exposure should be more properly defined as cases of unrecognized asbestos exposure [23,24]. Minerals not belonging to the asbestos group have also been shown to cause mesothelioma. This the case of erionite, which has been found in certain areas of Turkey. Erionite is a fibrous form of the zeolite group of minerals that has been shown to be several times more carcinogenic than crocidolite in the development of MPM. Until recently it was thought that the problem of exposure to erionite concerned only some Turkish villages; however, a report was lately published describing the first case pleural mesothelioma associated with erionite in the USA [25]. Several roads in North Dakota have been paved with materials containing erionite, explaining why the concentration of airborne erionite fibers in some areas is as high as or even higher than that found in Turkish villages with a corresponding increased incidence of erionite-associated MPM [26]. Tremolite present in northwestern Greece, although not extracted for commercial use, has often been found as a contaminant of chrysotile and is causally linked to an increased risk of developing mesothelioma. Moreover, a cluster of deaths from pleural mesothelioma has been reported in the Italian city of Biancavilla (Sicily). Subsequent studies have demonstrated that those MPM cases were related to the patient exposure to fluoro-edenite, a material extracted from quarries that features morphology and composition similar to that of minerals of the tremolite-actinolite series [14,15]. Similarly, pleural pathologic damages recalling MPM have been reported in animals exposed to carbon nanotubes [19]. Some evidence suggests that the exposure to chemicals as nitrosamines, nitrosureas, potassium bromate, and ferric saccharate could be a potential culprit in a context of genetic susceptibility [27,28,29,30]. Ionizing radiation, particularly for the treatment of a primary cancer, is known to increase the risk of MPM development. Notably, the MPM risk seems to be increased, even if the pleural layer has not been directly involved, and long latency periods are associated with higher relative risks, with a non-linear dose–response relationship [31].

## 4. Germline and Somatic Changes

Genetic and inherited factors play a significant role in cancer susceptibility. Genetic epidemiology is defined as the branch of epidemiology that studies the role of genetic factors, along with the environmental contributors to disease; in other words, it is focused on how and why diseases cluster in familial or other defined groups [32,33]. It is a relatively new research filed since it was first described in 1954 by Neel and Schull [34], resumed in the middle of the 1980s through linkage analysis, but takes major advances from the most recent genome-wide association studies. By integrating germline and somatic data, genetic epidemiology studies allow for the deciphering of the complex landscape of cancer and for associating certain genetic variants to specific phenotypes, for defining risk of disease onset, and for identifying novel actionable candidates.

### 4.1. Genetic Susceptibility Factors: Clues and Inferences

For any given level of exposure to a carcinogen, not all the exposed subjects develop the disease [35]. In this context, the threshold dose is defined as the point (dose) at which the effect (cancer) is first observed. On the other hand, individual cancer susceptibility might increase a person’s risk of disease onset and can be the result of several host factors. Indeed, few cancers are a result of mutations in a single gene, whereas more complex genetic traits intervene in most cases [36]. Concerning MPM, some point should be underlined. The first is the absence of a defined threshold and the long latency after exposure to biopersistent fibers. The latter defines a unique exposure–response relationship that reflects not a simple cumulative exposure to fibers but a more complicated temporal, biologic, and individual network. Moreover, MPM can occur spontaneously in absence of documented exposure to asbestos or other risk factors. Several relatively recent works have documented alterations in the BRCA1—associated protein gene (*BAP1*) in familiar MPM clusters. The discovery that germline *BAP1* mutations cause mesothelioma and other malignancies, namely, uveal cancer, meningioma, and melanoma, overall defined as “*BAP1*-related cancer syndrome”, elucidated some of the key pathogenic mechanisms [37]. The *BAP1* gene is located on chromosomal region 3p21.2–p21.31, a region that is frequently deleted in many other solid cancers, among which lung carcinomas. It consists of 17 exons (*BAP1* [BRCA1 associated protein 1 [Homo sapiens (human)]-Gene-NCBI: Pubs; 2016. http://www.ncbi.nlm.nih.gov/pubmed/ (accessed on 1 December 2020)). It encodes for a 90 kDA deubiquitinating enzyme (DUB), which regulates key cellular pathways, including cell cycle, cellular differentiation, transcription, and DNA damage response [38]. The gene is frequently lost by chromosomal deletion in various tumors. First report of *BAP1* germline mutations has been described in two families with high MPM incidence [39]. Germline mutation affecting BAP1 gene is inherited and exits in an autosomal dominant phenotype [40]. Similarly to other tumor suppressor genes, the affected subjects inherit a non-functional *BAP1* allele, whereas inactivation of the second allele occurs later in life. In animal models, *BAP1*^+/−^ mice exhibited significantly increased incidence and rate of progression of asbestos-induced MPM than wild type models [41]. Indeed, *BAP1*^+/−^ mice carried bi-allelic inactivation of the *BAP1* gene coherently to its predisposing role as a cancer susceptibility candidate gene. From comprehensive profiling of MPM high-risk families, *BAP1* germline mutations emerge as generally associated with less aggressive disease and a 3.5/7-fold improved survival, while in other cancer types, they are associated with poor prognosis. Moreover *BAP1*-mutated MPM most frequently feature epitheliod differentiation and predominant peritoneal involvement, and arise in female patients; the age of disease onset is overall lower than that in wild type patients and similar to that registered in other hereditary cancer syndromes [42]. This observed better outcome is certainly related to younger age of patients and to the more treatable peritoneal tumors [43,44]. Interestingly, in several families, there are individuals carrying mutated *BAP1* who harbor more than one type of primary cancer, and this observation strongly suggests that multiple tissue lineages are affected by *BAP1* deficiency [45,46,47]. *BAP1* loss has been consistently reported in approximately 50% of all sporadic mesotheliomas. In vivo studies have demonstrated that the occurrence of germline *BAP1* mutations in combination with asbestos exposure determine a significant increased incidence of MPM and more aggressive tumors, pointing out a synergy between genetic and environmental factors. It has been suggested that individuals carrying mutations in *BAP1* should avoid asbestos exposure, cigarette smoking, as well as excessive sun exposure. The identification of *BAP1* as a marker of familial cancer syndromes allowed for the design of *BAP1*-pharmacologic blockade strategies. Due to their role in homologous recombination repair [48], poly(ADP ribose) polymerase (PARP) inhibitors were tested for their efficacy in *BAP1*-_null_ tumor cells with contrasting results. In a first report, no differential sensitivity was observed between *BAP1*-_WT_ and *BAP1*-_mutant_ MPM cells treated with the MK4827 (Merck) inhibitor [49]. In another in vitro trial on chicken DT40 cells, increased sensitivity to olaparib was instead observed in homozygous *BAP1*-_null_ cells as compared to wild-type and heterozygous *BAP1*-_null_ cells [48]. More recently, it has been shown that significant sensitivity to PARP blockade might be related to the occurrence of an alternative splice variant of *BAP1*, which leads to the loss of 12 amino acids within the catalytic and BARD1 binding domains [50]. However, the use of PARP inhibition (olaparib, nirapanib) might induce synthetic lethality in MPM cells, irrespective of *BAP1* status [51]. In this context, it is known that PI3K inhibition impairs BRCA1 expression, and in fact treatment with olaparib in combination with the mTor inhibitor GDC0980 displays synergic effect in blocking MPM cell proliferation and survival [42]. Another relevant study demonstrated that *BAP1* can transcriptionally promote the expression of HDAC2. The *BAP1* knockdown led to a decrease in HDAC2 but an increase in HDAC1 expression. This imbalance seems to play a role in promoting MPM cell sensitivity to the histone deacetylase vorinostat [52]. Enhancer of zeste homolog 2 (EZH2) is upregulated in mesothelioma, and preclinical models have identified a possible association between *BAP1* loss and EZH2 upregulation. Specific EZH2 inhibitors (e.g., tazemetostat, EPZ-6438) decreased cell proliferation and reduced invasion and clonogenic capacity in mesothelioma cell lines and in in vivo experiments [53].

Although more rarely described, deleterious missense mutations in other genes have been found in familial clusters of mesotheliomas. The *CDNK2A* gene mutation is among the most relevant susceptibility mutation in patients affected by multiple primary melanomas and has been reported to be involved in mesothelioma as well. *CDKN2A*, also known as cyclin-dependent kinase inhibitor 2A, is located at chromosome 9, band *p*21.3, and encodes for two major proteins: p16(INK4), which is a cyclin-dependent kinase inhibitor, and *p*14(ARF), which binds the *p*53-stabilizing protein MDM2 (www.omim.org/entry/600160 (accessed on 1 December 2020)). Somatic *CDNK2A* mutations frequently occur in sporadic MPMs. However, in a report by Betti and colleagues, in two out of six families affected by both mesothelioma and melanoma, the authors showed *BAP1* germline nonsense mutation or a recurrent pathogenic germline mutation (c.301G > T, *p*Gly101Trp) in the *CDKN2A* gene [36,44]. Patients carrying these *CDNK2A* mutations were categorized as “low asbestos exposure”, without occupational exposure. This observation allowed the authors to hypothesize that *CDKN2A* germline mutations could lead to increased risk for developing MPM. Germline mutations in the *TP53* gene have also been reported. It is well known that germline *TP53* mutations are causatively associated with Li–Fraumeni syndrome [54]. Massive DNA sequencing and molecular inversion probe microarray analysis revealed germline *TP53* mutations associated with peritoneal malignant mesothelioma in women patients [55,56]. Furthermore, a *TP53* germline missense change (*p*Arg213Gln) was found in a large family characterized by the occurrence of different cancer types, among which MPM in a subject who did mention exposure to asbestos was found [57]. Finally, germline mutations in the *NF2* gene have been reported. It encodes for merlin, a tumor suppressor protein frequently inactivated in borderline tumors such as schwannoma and meningioma [58]. The sequence of merlin is similar to that of ezrin/radixin/moesin (ERM) proteins, which crosslink actin with the plasma membrane, thus indicating that merlin plays a role in transducing extracellular signals to the actin cytoskeleton [59]. However, it seems that a “second hit” somatic change is likely required to trigger the development of MPM [60]. Genome-wide association studies (GWAS) have confirmed those data and documented that other genes interact in the *BAP1* signaling pathway. For example, analysis of 407 pleural mesothelioma cases and 389 controls with a comprehensive history of asbestos exposure revealed an increased risk of abnormalities in chromosomal region 7p22.2, which includes the gene encoding the Forkhead box protein K1 (FOXK1) that interacts with *BAP1* [61]. However, it is critical to probe genes beyond *BAP1* that confer differential susceptibility to mesothelioma development and treatment response. Recent studies have identified novel susceptibility candidates. In Western Australia, a large study (428 cases and 1269 controls) reported variants located in the *CRTAM*, *SDK1*, and *RASGRF2* genes that were significantly associated with MPM risk [62]. In those cases, it is common to focus on the interactions between exposure to a specific environmental carcinogen (e.g., asbestos) and single-nucleotide polymorphisms (SNPs) emerging from previous GWAS studies. For instance, these approaches have revealed the synergistic effect between asbestos exposure and rs1508805, rs2501618, and rs5756444 genetic variations [63].

### 4.2. Somatic Events

Several acquired chromosomal alterations have been identified in MPM, including deletions in chromosome 3 (3*p*21), 9 (9*p*21), and 22 (22*q*12); number losses (4, 14, 18, 19, and 10); gains (5); or a combination (1, 8, 9, 11, 15, 16, 17, 21, and 22) [61]. Somatic point mutation affecting the *BAP1* gene located at 3p21.3 site have been reported in 20–25% of sporadic MPM cases [38,40,48], although subsequent studies applying next-generation sequencing detected higher incidences (60%) of bi-allelic homozygous deletions in sporadic MPM cases [62,63,64]. Truncating mutations and aberrant *BAP1* expression were common in sporadic MPMs, in the absence of germline changes. This evidence confirms that *BAP1* behaves as a tumor suppressor. Interestingly, the occurrence of *BAP1* mutations/deletions seems to be an early event in MPM onset as demonstrated by extensive whole-exome sequencing analysis of in situ MPMS [65]; they are associated with a prolonged progression free survival for patients treated with platinum/pemetrexed regimens [66]. Massive genetic studies revealed recurrent somatic mutations in other tumor suppressor genes, such as *CDKN2A* and *NF2*. Exon sequencing analysis on platinum/pemetrexed-treated MPM revealed somatic changes in 23 genes including *BAP1*, *TP53*, *NRAS*, and *EGFR*, as well as increased copy number of the *CDKN2A* and *CDKN2B* genes [67]. Somatic changes were also documented in the cullin 1 gene (*CUL1*), phosphatidylinositol-4-phosphate 3-kinase catalytic subunit type 2 beta gene (*PIK3C2B*), TAO kinase 1 gene (*TAOK1*), and radixin gene (*RDX*) [60]; changes in *RAPGEF6* and *ACTG1* genes are mainly reported in non-epithelioid subtypes [65]. Overall, the most relevant consideration popping out from the different studies is the absence of somatic activation in a confirmed driver gene. This weak mutational landscape of MPM suggests two relevant epidemiologic implications. The first is that it is more likely that several mutations need to accumulate before MPM development. This point is coherent with the long latency of MPM onset after exposure to asbestos. Secondly, this condition is a consequence of the selective pressure played by the microenvironment, mainly by the inflammatory circuits locally triggered by bio-persistent fibers [68]. Indeed in vitro and in vivo experiments showed that asbestos induces inflammation before the development of tumor. Notably, human mesothelioma cells and macrophages upon asbestos exposure release the high mobility group box 1 (HMGB1) protein that is a key player in triggering inflammatory cascade [69] and autophagy, thus directly promoting cell survival and malignant transformation [70]. The proinflammatory molecule HMGB1 is emerging as potential biomarker and therapeutic target in MPM [71]. Upon asbestos-induced chronic inflammation, the HGBM1 is secreted from mesothelial cell into the stroma and after it can be found in systemic blood circulation [69]. Within respect to the epidemiologic perspective, serologic levels of HMGB1 have been validated as predictive biomarker for monitoring occupational workers and their families who have a history of exposure to asbestos [72].

### 4.3. Gene Signatures in MPM: What We Know and What Is Still Missing

The occurrence of somatic changes in MPM genome that, as showed above, are caused by multiple mutational processes, generates characteristic mutational signatures [73]. Among those signatures associated to DNA mutations, one is related to the production of reactive oxygen species (ROS). The advances in experimental modelling of mutational signatures are contributing, on one hand, to the elucidation of the role of genetic profiling in MPM diagnosis and classification, and, on the other, to facilitating the design of innovative trials and more specific therapeutic approaches. The PI3K/Akt signaling pathway is involved in cell survival and its activity can be redox-regulated. This point is particularly relevant in MPM, since it is well known that asbestos toxicity is mainly related to the release of ROS [74,75]. The analysis of transcriptomes and whole exomes confirmed that the PI3K/mTor pathways are activated in MPM [76]. In detail, it has been reported that this activation is associated to the occurrence of mutation in the neurofibromatosis 2 (*NF-2*) gene. The latter acts as tumor suppressor that is part of the *NF-2*/Merlin complex that makes up the *NF2*/Hippo pathway. Overall, patients with *NF-2* mutation had a worse prognosis [77]. Recent data have linked *NF-2* and *BAP1* to ferroptosis, a type of programmed cell death dependent on iron and characterized by the accumulation of lipid peroxides [78,79]. In ferroptosis, cell death is mediated by ROS, which induces peroxidation of polyunsaturated fatty acids (PUFAs). In MPM, the abundance of PUFAs is dependent on the level of acyl-CoA synthetase long-chain family member 4 (ACSL4) expression, which is activated by YAP/TAZ, while *BAP1* inhibits the expression of the scavenger SLC7A11 [80]. Thus, activating mutations affecting the NF2-YAP signaling can be therapeutically targeted by ferroptosis-inducing agents [78]. Further next-generation analysis has allowed the identification of signatures related to gender and neoplastic morphology. Epithelioid MPM forms are associated with increased incidence of *BAP1* single nucleotide variants and more frequent loss of chromosome 22 q. The non-epitheliod subtypes carry more frequent *CDKN2A* deletions in men. Overall, in the analyzed cohort, the loss of *CDKN2A* is associated with shorter survival and in male patients. Otherwise, TP53 mutations have been detected more frequently in females [81]. Interestingly, the investigation of recurrent gene alteration in the four MPM clusters identified by RNA-seq (RNA sequencing) expression performed by Bueno and colleagues [76] allowing for the identification of specific gene signatures in each cluster. The authors detected four genes (*SETD2*, *TP53*, *NF2*, and *ULK2*) as the most significantly mutated genes featuring significant differences among cluster 1 (sarcomatoid MPM) and cluster 2–4 (epitheliod, biphasic-epithelioid, and biphasic-sarcomaitoid MPM), thus suggesting that the differences observed in overall survival in cluster 1 might be related to the differential combination of gene alterations. Busacca et al. demonstrated on tissue samples differential miRNA signatures between biphasic-sarcomatoid and epithelioid-sarcomatoid tumors, whereas no differences in terms of histotypes were found with regards to the expression of hsa-miR-31, hsa-miR-221, or hsa-miR-222 [82]. Among that which is still missing for gene signature, no attempt thus far has been reported in terms of applying Imaging Genomics in this setting, again underlining MPM as being an orphan disease. Only two studies on radiomics of MPM have been reported, which focused on plaques characterization (diagnosis) and outcome prediction [83,84]; no integrations between imaging features and genetic characteristics have been attempted.

## 5. Gene Expression Modulators

Information on gene expression and modification are gaining increasing relevance in molecular epidemiology studies since it reflects both genetic and environmental influences [85]. Epigenetic changes are significantly associated with asbestos burden and strongly predict clinical outcome [86]. MicroRNAs (miRNAs) are endogenous, evolutionarily conserved, small non-coding RNAs that function in regulation of gene expression [87]. Several miRNAs are known to play a role in carcinogenesis and can also function as therapeutic targets. The relationship between epigenetic and miRNAs is still poorly known. miRNAs can be involved in DNA methylation and can directly target enzymatic effectors of the epigenetic machinery [88]. Similarly, siRNAS (small interference RNAs), carrying a structure closely related to that of miRNAs, can modulate DNA methylation and histone modifications. On the other hand, a number of therapeutic agents targeting tumor epigenetic assett also act by affecting miRNA expression [89]. Growing evidence has reported that non-mutagenic alterations are common events in MPM onset and progression. Coherently, the rate of tumor mutational burden is low when compared to other solid tumors [90]. Data on microsatellite instability (MSI) in MPM are poor and still non conclusive. A large retrospective analysis on MPM tissue samples did not report any case of MSI, although the study was performed only with immunohistochemistry [91]. Thus, different mechanisms of gene expression are emerging as of potential utility in the molecular epidemiology field.

### 5.1. Epigenetic Alterations

Epigenetics refers to the heritable changes beyond the DNA sequence, which are reversible and include different processes such as DNA methylation, chromatin modifications, nucleosome positioning, and alterations in non-coding RNA profiles. Epigenetic mechanisms modulate gene expression and thus influence disease onset and progression. For this reason, it results in a focus of molecular epidemiologic investigation [92]. To this purpose, it should be remarked that the epigenetic state of genes is dynamic and subjected to fluctuations [93]. Aberrations in the normal DNA methylation patterns and histone modifications have been recognized as targets for cancer therapy [94]. Epigenetic alterations in MPM frequently occur in disease onset and progression [95,96]. A comparison of epigenetic profiles of a vast series of MPM tissue samples has documented an average number of 387 hypermetilated genes. A specific epigenetic signature has been proposed (encompassing the *TMEM30B*, *KAZALD1*, and *MAPK13* methylated genes) and has been suggested to be predictively associated to MPM development [97]. The epigenetic silencing of *SFRP2* and *SFRP5* tumor suppressor genes, which encode secreted frizzled-related proteins, has been observed in MPM samples. Promoter methylation profile of these genes could potentially serve as a plasma-based epigenetic biomarker [98]. Moreover, SFRPs are antagonistic modulators of the Wnt signaling, and their downregulation via hypermethylation could be related to the aberrant activation of the Wnt pathway that has been reported in MPM [99]. Moreover, a lower frequency of DNA hypermethylation has significantly correlated with a better clinical outcome. Indeed, experimental findings on cultures and animals showed that asbestos induces inflammation before the development of a tumor. Notably, human mesothelioma cells and macrophages upon asbestos exposure release the high mobility group box 1 (HMGB1) protein that is a key player in triggering inflammatory cascade [69] and autophagy, thus directly promoting mesothelial cells survival and malignant transformation [70]. HGBM1 acts as a nonhistone chromatin-binding protein that targets DNA and drives transcription factor assembly [100]. Subsequently, HGBM1 can be measured on systemic circulation, and its serum level has been validated as both a prognosis and exposure marker [101]. Notably, increases expression of these genes significantly affect patient outcomes [95]. Other dynamic mechanisms of epigenetic modifications involve the acetylation and methylation of histones, as well as chromatin remodeling. The study by Goto et al. showed that several genes silenced in MPM samples were enriched for the repressive histone H3 lysine methylation (H3K27me3) mark [100], which is a downstream target of PRC2 (the polycomb repressive complex 2). Experimental evidence further indicated that increased H3K27me3 marks were associated with concomitant overexpression of EZH2 (enhancer of zeste homolog 2) and SUZ12 (SUZ12 polycomb repressive complex 2 subunit) that, in forming the PRC2 complex, contribute to tri-methylation of histone 3 lysine. Moreover, the increased expression of EZH2 and SUZ12 correlated with poor survival, highlighting the potential of therapeutic targeting of these deregulations in MPM [102]. In addition, Lafave et al. demonstrated that the loss of *BAP1* in mice resulted in increased H3K27me3 and elevated expression of EZH2. Furthermore, pharmacological inhibition of EZH2 in *BAP1*-mutant MPM cell lines blocked their growth, and findings in preclinical models have suggested a relationship between *BAP1* and activation of EZH2 [103]. In strong support of these findings, tazemostat, a first-in-class small-molecule inhibitor of EZH2, received accelerated FDA (Food and Drug administration) approval in January 2020 for the treatment of locally advanced or metastatic epithelioid sarcoma [104]. More importantly, it is currently undergoing clinical development for use in other tumor types, including MPM. Epigenetic therapy is a growing area of focus for cancer research and therapy. This strategy is mainly relevant in MPM management since this tumor rarely harbors those somatic genetic drivers. Mesothelioma exhibits the silencing of tumor suppressor genes through methylation and deregulation of polycomb group (PcG) proteins, as well as loss of imprinting (LOI) and de-repression of CG islands. DNA methyltransferases (DNMTs) are potentially attractive targets due to their role in silencing tumor suppressors. However, the available clinical data regarding DNMT pharmacological inhibition are still disappointing [104,105].

### 5.2. Diagnostic and Prognostic Role of miRNAs

From the different cell types, miRNAs can be secreted into the extracellular space and then transported to the circulating blood flow and in body fluids as pleural effusions [106]. The vast majority of scientific efforts within the study of miRNA expression in MPM have been directed to the identification of differential signatures between transformed and inflammatory mesothelium [107,108,109] in order to predict the risk of disease development that is based on certain exposure to asbestos [110,111] and to classify MPM subtypes [112]. Many experimental reports have identified members of the oncomiRNA *miR 17-92* cluster and its paralogs, namely, *miR 17-5p*, *18a*, *19b*, *20a*, *20b*, *25*, *92*, *106a*, and *106b* to be markedly upregulated in MPM if compared to normal mesothelial cell lines [113]. Moreover, lower expression of *let-7e*, *miR-7-1*, *miR-9*, *miR-34a*, *miR-144*, *miR-203*, *miR-340*, *miR-423*, and miR-582, and a higher expression of *let-7b*, *miR-30b*, *miR-32*, *miR-195*, *miR-345*, *miR-483-3p*, *miR-584*, *miR-595*, *miR-615-3p*, and *miR-1228* have been detected in MPM tissue samples compared with normal cells [107]. *miR-101-3p* and *miR-494* have been found to be downregulated in chronic pleuritis and pleural hyperplasia, whereas the same study reported *miR-181a-5p*, *miR-101-3p*, *miR-145-5p*, and *miR-212-3p* expression to be reduced in transformed tissues [114]. The level of circulating *miR-103* have been reported to be significantly decreased in MPM patients vs. health subjects [115]. The majority of these miRNAs were either located in chromosomal areas generally known as aberrant in MPM or were targeting well-described genes involved in MPM tumorigenesis. Over-expressed *miR-30b*, *miR-32*, *miR-483-3p*, *miR-584*, and *miR-885-3p* target tumor-suppressor genes such as *CDKN2A*, *RB1*, and *NF2*. The overexpression of *miR-29c* has been reported to behave as a tumor suppressor and mediates epigenetic effects, leading to downregulation of *DNMT1* and *DNMT3A* genes [116]. For instance, downregulation of *miR-31* expression is related to the deletion of the *miR-31* gene located in chromosome 9p21.3, which is commonly altered in aggressive MPMs. Indeed *miR-31* loss is associated the development of both chemo- and radioresistance in MPM cell lines [117]. On the other hand, down-regulated *miR-9*, *miR-7-1*, and *miR-203* target *EGF*, *HGF*, *JUN*, and *PDGFA* oncogenes. Interestingly, miRNa expression has been tested in pleural effusion as well. This approach is of particular interest for MPM with potential prognostic and therapeutic implications. In 2019, Birnie and colleagues investigated the microRNA profile of neoplastic pleural effusion compared with inflammatory one and found that the combination of *miR-143*, *miR-210*, and *miR-200c* could significantly differentiate MPM [118]. Predictive role of circulating miRNA signatures has been tested in asbestos-exposed subjects. Matboli et al. analyzed the serum microRNA profile of MPM with four public microRNA databases (miR2Disease, miRWalk, Human MiR, and Disease Database) and reported that *miR**-548a**-3p* and *miR**-20a* were potential diagnostic markers for MPM [119]. The authors also discovered that serum hsa-miR-2053 behaves as an independent prognostic factor of MPM [120]. Malignant cells can also secrete exososomes, microvescicles that could contain also nucleic acids. Serum exosomal miRNa signatures have been proposed as diagnostic markers of MPM. Cancer cells can secrete exosomes into the circulation, and the signature of proteins and nucleic acids contained in exosomes are significantly correlated with those in primary tumor cells. Therefore, serum exosomal microRNA has been proposed to be a potential diagnostic marker for MPM. For instance, MPM cells secrete high levels of *miR-16-5p*, *miR**-103*, *miR**-98*, *miR**-148b*, *miR**-744*, and *miR**-30e**-3p*, and these signatures present diagnostic accuracy and can be also exploited for therapeutic purposes [121,122]. Comparative analysis of the *miR-31* expression in MPM and reactive mesothelial proliferations has demonstrated that it is significantly upregulated in transformed mesothelioma, specifically in sarcomatoid subtypes [123]. Differential signatures, although in a limited cohort of samples, have reported upregulation of *miR101*, *miR25*, *miR26b*, *miR335* and *miR433*; two miRNAs were found to be downregulated (*miR191*, *miR223*), and two miRNAs were expressed exclusively in patients (*miR29a* and *miR516*) [124]. Polymorphisms in the miRNA pathway (miR-polymorphisms) were first noted in 2006 and now are emerging as powerful tools to decipher cancer biology. Indeed miRNA-associated single nucleotide polymorphisms (SNPs) might interfere with the function of miRNAs themselves and, consequently, to gene expression and signal transductions [125]. Several studies demonstrate that the analysis of miRNA-related variants can predict risk of cancer development but correlate with therapeutic and clinical outcome. They essentially represent another type of genetic variability that can affect cancer evolution [126]. Different computational case–control approaches can be used to identify those SNP candidate variants that deserve functional characterization. Up until now, few data are available on miRNA polymorphisms in MPM. In a cohort of asbestos-exposed subjects, detection of *miRN-146A rs2910164* polymorphism has been significantly associated with risk of MPM onset since carriers of two polymorphic alleles features lower risk of disease development [127]. Within an extensive occupational surveillance program on workers previously exposed to asbestos, researchers documented, with potential diagnostic and prognostic implications, that the rs1057147 variant within the 3′UTR of the mesothelin (*MSLN*) gene could be a putative miRSNP and that MSLN is a candidate target of miR-611 and, to a lesser extent, of miR-887 [128].

## 6. Cutting-Edge Molecularly Oriented Therapies

The recent improvement in the knowledge of molecular mechanisms leading to pleural malignant transformation has allowed for the start of novel and more promising tailored therapeutic approaches [67]. MPM is essentially resistant to radiotherapy. Overall, discouraging results from conventional chemotherapeutic approach, together with disease incidence data, has led MPM to be listed an orphan disease by the European Union (http://www.rarecarenet.eu/ (accessed on 1 December 2020)). That is why in MPM there is a strong rationale to exploit molecular epidemiology in discovering and validating potentially actionable markers. Novel therapeutic approaches are summarized in Figure 1B. Ideally, the identification of disease molecular markers could be of help in a more proper tumor histological subtype classification as well as in a more efficient patient stratification towards targeted agents.

### 6.1. Gene Therapy

To date, the management of MPM is still complex, and patient survival remains poor. However, most promising therapeutic programs are emerging on the basis of the more recent advances in molecular characterization and evolution of the disease. Although a detailed description of all the novel therapeutic approaches goes beyond the scope of this paper (for a review, see [67,129,130,131]), a brief synopsis of those approaches that have taken advantage in their development towards the clinical arena from the evolution of a molecularly based approach is necessary. In this perspective, gene therapy emerges among the most interesting ways to improve MPM outcomes. It is essentially aimed at replacing or repairing mutated genes that were harmed after asbestos exposure. Notably, because of the anatomy features (thin mesothelial layer made of cells with large surface) that efficiently act in gene transfer, MPM identifies a good target for gene therapy. Different viral and non-viral vectors (liposomes, bacteria, non-replicating viruses) have been engineered [132,133]. Technically the vector is inserted into the pleural space and binds to cancer cells. Afterwards, a specific gene is introduced into the cancer cells. Notably, MPM rarely metastasizes to distant sites, and the constant ventilation movements allow for the distribution of the vector within the pleural space. Several clinical trials with adenovirus (Ad) vectors have been shown to be safe and feasible. The identified targets are represented by (i) MPM gene defects. The most frequent homologous deletion concerns the INK4A/ARF locus encompassing the *p14ARF* and the *p16INK4A* genes. The resulting genetic defects are defined by inactivation of the p53-mediated pathways and progression of cell cycle. Reintroduction of defective gene should be performed through an adenoviral vector bearing the *p14* or the *p16* gene (Ad-p14, Ad-p16), suppresses the cell proliferation through pRb dephosphorylation, and reactivates the p53-mediated checkpoint mechanisms. (ii) Gene therapy targeting signal transduction. A main interesting example is related to the observation that MPM is characterized by overexpression of VEGF and VEGFR1 and two pathways, as well as the HGF–MET pair [134]. The NK4 molecule is a competitive antagonist of HGF, designed to impair the MET-driven invasive growth program [135]. Notably NK4 also inhibits VEGF-induced neo-angiogenesis since it indirectly suppresses the VEGF-mediated angiogenic environments [136]. Thus, it emerges among the most potentially effective agents for gene therapy, and several preclinical sources of data have shown anti-neoplastic activity of adenovirus expression in the *NK4* gene [137]. (iii) Oncolyitc virotherapy. This defines a therapeutic strategy that uses viruses to target and kill cancer cells. Several oncolytic viruses have been texted in MPM, such as Adenoviruses, measles, retroviruses, Newcastle disease, herpes simplex viruses, vesicular stomatitis virus, and Sendai viruses. An interesting example is related to the fact that some MPMs express the coxachie adenovirus receptor (CAR) molecules, the major type 5 Ad receptors, at a low level. The substitution of the fiber-knob region, which mediates a binding to CAR with that of type 35 Ad, improves the infectivity to mesothelioma since CD46, the major type 35 Ad receptor, is expressed at a high level in several MPM cell lines. Notably CD46 is selectively expressed in malignant cells and not in inflammatory mesothelium, thus allowing for a highly specific delivery [138]. Overall combinations of oncolytic viruses with other types of cancer treatments results in synergistic effects [139]. Virotherapy, in combination with immunotherapy in particular, has shown promising results in MPM. An oncolytic adenovirus coding for human granulocyte–macrophage colony-stimulating factor showed immune activation abilities in a phase 1 trial on different types of solid tumors, even though only one of the two patients with MPM included showed disease stabilization, while the other patient exhibited progressive disease [140]. It has been shown that VSV-mIFNβ encoding the murine *interferon-β* gene can activate CD 8 T cells against MPM when delivered locally into the pleural space. Another study showed that MV is an appropriate vector for immunotherapy when used in combination with anti-PD-L1 or anti-CTLA-4 antibodies, therefore having potential application in MPM therapy [141].

### 6.2. Targeting Epigenetic Damages

The lack of efficacy of DNA hypomethylating agents in solid tumors has been related to their high systemic toxicity, mainly myelosuppression. Clinical efficacy of vorinostat, a histone deacetylase inhibitor, has been investigated in a phase 3 trial, as a second-line or third-line therapy improved patients’ outcome. Quite surprisingly, vorinostat did not improve OS (Overall survival) and could not be recommended as a therapy for patients with advanced cases [142]. Very interestingly, on the basis of the observation that HDAC inhibitors seem to play immunomodulatory effects [143], great interest has now been focused upon the combined use of these agents and immune checkpoint inhibitors or adoptive cell transfer strategies. In in vitro experiments on MPM cell lines, decitabine, and HDAC inhibitor induced a slight increase of the PD-L1 expression, and these data strongly suggest the rationale for a combinatorial approach between the two agents and anti PD-L1 molecule [144]. The tumor suppressors *BAP1* and ASXL1 interact to form a polycomb deubiquitinase complex that removes monoubiquitin from histone H2A lysine 119 (H2AK119Ub). It has been demonstrated that *BAP1* loss is associated with increased trimethylated histone H3 lysine 27 (H3K27me3), elevated enhancer of zeste 2 polycomb repressive complex 2 subunit (Ezh2) expression, and enhanced repression of polycomb repressive complex 2 (PRC2) targets [103]. Recent data have emerged on the safety and efficacy of tazemetostat (TAZ), a potent and selective EZH2 inhibitor, in relapsed/refractory (R/R) malignant mesothelioma with *BAP1*-inactivationevaluated in an open-label, phase 2 trial. Overall TAZ showed antitumor activity in *BAP-1*-eficient MPM patients and was generally well tolerated [145]. Moreover, *BAP1* stabilizes BRCA-1 and promotes poly-(ADP-ribose)-dependent recruitment of polycomb deubiquitylase complex PR–DUB to DNA damage sites [146]. *BAP1* isoform lacking part of the catalytic domain sensitized MPM cells to the PARP1 inhibitor, olaparid, and this sensitivity could be augmented by concomitant treatment with apitolisib (GDC-0980), a dual inhibitor of class 1 PI3K, and mTOR kinase, which downregulates BRCA-1.

### 6.3. Antiproliferative Effects of miRNAs

miRNA pharmacogenomics can be defined as the study of miRNAs and polymorphisms affecting miRNA function in order to predict drug behavior and to improve drug efficacy. A growing body of literature is emerging related to the potential application of microRNA-based therapy. Preclinical studies have pointed out that re-expression of miR-145 [147], miR-126 [148], and miR-34 [149] when transfected into MPM cells can significantly impact on tumor growth. Only one study has been available up until now in clinical settings with regards to the use of TargomiRs technology, namely, targeted minicells containing a microRNA mimic [150]. They are made of three components: (i) a miR-16-based microRNA mimicking a double-stranded, 23-base pair, synthetic RNA molecule; (ii) biologically engineered drug delivery vehicle EnGeneIC Dream Vectors (EDVs), which are nonliving bacterial minicells (nanoparticles) that function as leak resistant micro-reservoir carriers that allow for efficient packaging of a range of different drugs, proteins, or nucleic acids; and (iii) targeting specificity, wherein the EDVs are targeted to EGFR-expressing cancer cells with an anti-EGFR bispecific antibody. A phase I study, MesomiR-1, was designed to assess safety and activity of EDV targomiRs for patients with MPM and advanced failing on standard therapy (NCT02369198, [151]). This approach has shown acceptable safety profile and early signs of activity of targomiRs in those subsets of patients in combination with chemotherapy or immune checkpoint inhibitors [152,153], although careful measurement in terms of pre–post biopsies should be required to more deeply elucidate the therapeutic efficacy thresholds.

## 7. Evolving Technologies and Future Perspectives: From Reactive Medicine to Precision Medicine

As well what has been discussed above, MPM also represents an enduring challenge for physicians. Both the diagnostic process and the therapeutic protocols have reached high levels of complexity by integrating innovative variables from complex -*omics* domains, shifting the clinical paradigm towards real personalized and preventive healthcare. Therefore, besides the traditional clinical variables (e.g., age, sex, histology, TNM (Tumor Node Metastases) stage), new -*omics* entities, such as radiomic or radiogenomic signatures, will probably be introduced in clinical practice in the near future, allowing for the establishment of revolutionary treatments that pave the way towards a better care in mesothelioma patients. However, technologic improvements have brought about the biological and clinical complexity of this disease when facing the overwhelming number of variables to be consider in prevention, diagnosis, and treatment decisional processes. This traditional approach (the so-called “reactive medicine”) is quickly reaching its limits and also becoming economically unsustainable.

This ongoing revolution has also precipitated concerted efforts by many investigators to define molecular subgroups of mesothelioma, characterize the genomic landscape of different subtypes, identify novel therapeutic targets, and define mechanisms of sensitivity and resistance to targeted therapies. Therefore, besides histological, clinical, and demographic information, a wide range of data obtained from genomics, proteomics, immunohistochemistry, and imaging should be integrated by physicians to improve patient survival on the basis of more effective stratification tools, namely, a precision medicine approach.

Such a multiomic approach will provide physicians with an enormous amount of data (“big data”). Apart from analyzing and interpreting them individually, the task to integrate these rapidly expanding non-structured data will soon become too complicated for one medical doctor.

In this framework, with the advent of artificial intelligence (AI), nowadays physicians potentially have the adequate technology to deal with precision medicine approaches. Indeed, AI excels in recognizing complex patterns in medical data and provides a quantitative, rather than qualitative, assessment of clinical conditions. AI methods are precise and allow for specific quantification of features not otherwise quantifiable by human experts. The growing complexity of the human–machine and human–software interactions in conjunction with the increasing incidences of cancer have created a workforce shortage throughout the world. Furthermore, the knowledge and experience gap between more developed and under-resourced healthcare environments poses an enormous public health challenge and represents one of the great global inequities in cancer care.

## 8. Conclusions

A multidisciplinary approach to MPM has allowed for the development of molecular and genetic epidemiology. This approach is emerging as a powerful tool for this rare and aggressive disease for diagnostic, prognostic, and therapeutic purposes. Although there is a strong interconnection among different expertise and know-how across clinics, basic science, biostatistics, and bioinformatics are required to properly collect and process samples; to analyze and validate biomarkers; as well as to design large-scale screening, monitoring, and risk studies. The application of -*omics* techniques to MPM is at the beginning, and limitations are related to its rarity, the nonspecific symptoms, and the long latency of arousal after asbestos exposure. Otherwise, it could represent an exciting strategy to improve the knowledge on pathogenic mechanisms on one hand, and on novel therapeutic agents and tools on the other.

## Figures and Tables

**Figure 1 jcm-10-01034-f001:**
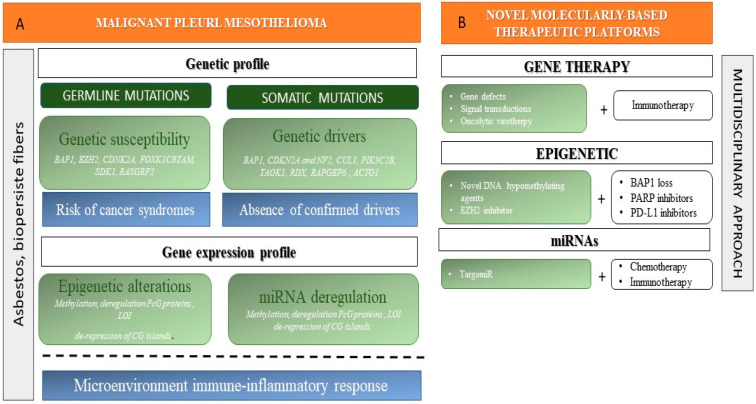
**Malignant pleural mesothelioma (MPM): Pathogenic mechanisms and therapeutic platforms**. (**A**) Main players involved in MPM onset. (**B**) Novel therapeutic strategies designed on the basis of disease molecular features.

## Data Availability

Not applicable.

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
