# Peer review of "The Evolving Landscape of the Molecular Epidemiology of Malignant Pleural Mesothelioma"

_jcm, 2021, doi:10.3390/jcm10051034_

Round 1

Reviewer 1 Report

This paper reviews a multidisciplinary approach for the molecular epidemiology of MPM. MPM is a rare and aggressive malignant tumor that originates from the pleura due to mainly asbestos exposure. MPM is increased worldwide despite being banned from using asbestos in many countries. It is an urgent issue how to screen and treat MPM patients. Various biomarkers have been studied for MPM. With technological progress, it is expected that comprehensive molecular epidemiology will advance the elucidation of the pathogenic mechanisms of MPM.

Comments:

In "4.2. Somatic events" (line 353-), HMGB1 does not appear to be relevant to this chapter (it is also described in line 505-).

There is an overlap (or similar sentence) of text and citations (line 391-417, line 582-585).

As the authors mentioned, MPM has some histologic subtypes (mainly epithelioid, biphasic, sarcomatoid-type), the prognosis of each subtype is different. I think it would be better to add some knowledge about the subtypes in the introduction.

It seems the letters in the light green box ("Genetic susceptibility", "Genetic Drivers", and so on) of Figure 1 are difficult to read in white. It would be better to reconsider the color or the design.

Small point:
Please check below.
line 97: "ae" in "aetiology" is italicized.
line 311, 317, 321: Is CD"NK"2A a mistake of CDKN2A?
line 460: citation [96, 97, y, z?]
line 619: radiotherap"y"?

Author Response

Reply to Reviewer 1

Comments and Suggestions for Authors

This paper reviews a multidisciplinary approach for the molecular epidemiology of MPM. MPM is a rare and aggressive malignant tumor that originates from the pleura due to mainly asbestos exposure. MPM is increased worldwide despite being banned from using asbestos in many countries. It is an urgent issue how to screen and treat MPM patients. Various biomarkers have been studied for MPM. With technological progress, it is expected that comprehensive molecular epidemiology will advance the elucidation of the pathogenic mechanisms of MPM.

We really thank the Reviewer for comments and for suggestions which improve the quality of the manuscript. We have revised and updated the manuscript. Below the point-by-point answers (A).

Major comments

  1. In "4.2. Somatic events" (line 353-), HMGB1 does not appear to be relevant to this chapter (it is also described in line 505-).

A1. We removed the reference as indicated

  1. There is an overlap (or similar sentence) of text and citations (line 391-417, line 582-585).

A2. We really thank the Reviewer for careful revision of the text. We removed similarities as suggested.

  1. As the authors mentioned, MPM has some histologic subtypes (mainly epithelioid, biphasic, sarcomatoid-type), the prognosis of each subtype is different. I think it would be better to add some knowledge about the subtypes in the introduction.

A3. We thank the Reviewer for this fruitful comment. The text has been implemented as follows:” …. Prognosis is extremely poor, with a median survival of about one year from diagnosis [ ], and depends mainly on  histological subtype. MPM comprises three main histological variants: epithelioid (60-80% of cases), sarcomatoid (10%) and biphasic/mixed (10-15%) [ ]. Epithelioid MPM shows proliferation of oval, cuboid or polygonal cells, is associated to better prognosis and is the only variant where surgery can prolong survival [ ] ; sarcomatoid subtype is characterized by the proliferation of spindle tumor cells and has the worst prognosis , whereas biphasic variant, which includes  both epithelioid and sarcomatoid elements, each represented for at least 10% of the neoplasm, carries an intermediate prognosis[ ]. 

  1. It seems the letters in the light green box ("Genetic susceptibility", "Genetic Drivers", and so on) of Figure 1 are difficult to read in white. It would be better to reconsider the color or the design.

A4. We thank the Reviewer for the suggestion. We modified the color to make the figure easy to read.

Small points:

Please check below.
line 97: "ae" in "aetiology" is italicized.
line 311, 317, 321: Is CD"NK"2A a mistake of CDKN2A?
line 460: citation [96, 97, y, z?]
line 619: radiotherap"y"?

Answer to minor comments: we really thank the Reviewer for the careful revision of the manuscript. All the points have been revised.

Reviewer 2 Report

Molecular epidemiology increases our knowledge about the pathogenesis of disease by identifying vital molecules, genes and pathways that have effect on the risk of disease development. This review article is an excellent introduction to the molecular epidemiology in malignant mesothelioma and presence a new opportunity for the medical community to understand accurate molecular base of disease such as mesothelioma.

The review paper is well written, concise and accurate. It flows very well and makes reading this review article clear and easy to follow. To my knowledge there has not been any review articles looking at molecular epidemiology in malignant mesothelioma. The article would be of great interest to medical oncologist, pulmonologist, thoracic surgeons as well as medical scientist in the field of mesothelioma.

At present, I have no specific changes to the content. There are some minor grammatical and spelling errors through the manuscript. The conclusion is well written and consistent with the evidence provided.

Author Response

Reply to Reviewer 2

Comments and Suggestions for Authors

Molecular epidemiology increases our knowledge about the pathogenesis of disease by identifying vital molecules, genes and pathways that have effect on the risk of disease development. This review article is an excellent introduction to the molecular epidemiology in malignant mesothelioma and presence a new opportunity for the medical community to understand accurate molecular base of disease such as mesothelioma.

The review paper is well written, concise and accurate. It flows very well and makes reading this review article clear and easy to follow. To my knowledge there has not been any review articles looking at molecular epidemiology in malignant mesothelioma. The article would be of great interest to medical oncologist, pulmonologist, thoracic surgeons as well as medical scientist in the field of mesothelioma.

At present, I have no specific changes to the content. There are some minor grammatical and spelling errors through the manuscript. The conclusion is well written and consistent with the evidence provided.

We really thank the Reviewer for these positive comments and careful review, which helped improve the manuscript. The grammatical and typo errors have been revised.

Reviewer 3 Report

The work "The evolving landscape of the molecular epidemiology of malignant pleural mesothelioma" by Lettieri et al. is a comprehensive review and well written.

No major revisions are needed. However, some minor points need revision.

Line: 61-62: "...was not definitively proven." sounds like SV40 might still be a plausible cause for mesothelioma. However, several studies showed a contrary position, whereas the reference is 12 years old. Additionally, any hypothetical SV40 connection is not relevant for this review.

Line 120-126: The selected limitations are presented unclear, e.g. iii) and iv) might be important for longitudinal studies, but not for initial case-control studies. Additional, regarding v) samples from several studies are used in common biomarker studies. Thus inter-individual variations could be observed. Thus, the presented limitations should be presented in more detail.

Line 218-219: The reference of the sentence "Ionizing radiation, particularly for the treatment of a primary cancer, is known to increase the risk of MPM development." is missing.

Line 592-594: There are some typos in the miRNAs.

Author Response

Reply to Reviewer 3

Comments and Suggestions for Authors

The work "The evolving landscape of the molecular epidemiology of malignant pleural mesothelioma" by Lettieri et al. is a comprehensive review and well written. No major revisions are needed. However, some minor points need revision.

We really thank the Reviewer for the comment. We have revised the minor points, as suggested. Below the point-by-point answers (A).

  1. Line: 61-62: "...was not definitively proven." sounds like SV40 might still be a plausible cause for mesothelioma. However, several studies showed a contrary position, whereas the reference is 12 years old. Additionally, any hypothetical SV40 connection is not relevant for this review.

A1. We agree with this comment and the text has been modified as follows: “… Simian Virus 40 (SV 40) was previously explored as etiologic agent, but its role wasn’t subsequently proven”.

  1. Line 120-126: The selected limitations are presented unclear, e.g. iii) and iv) might be important for longitudinal studies, but not for initial case-control studies. Additional, regarding v) samples from several studies are used in common biomarker studies. Thus inter-individual variations could be observed. Thus, the presented limitations should be presented in more detail.

A2. We thank the Reviewer for pointing out this critical issue. The text has been modified according to Reviewer suggestions, as follows: “ …For instance, laboratory errors can lead to biomarker misclassification. In this perspective, dedicated studies should be performed to assess the degree of reliability…. These points might be important for longitudinal studies, but not for initial case-control studies. …Samples from several studies are used in common biomarker studies. Thus, inter-individual variations could be observed..”

  1. Line 218-219: The reference of the sentence "Ionizing radiation, particularly for the treatment of a primary cancer, is known to increase the risk of MPM development." is missing.

A3. We agree with this suggestion and we introduced the following reference: “ Gilbert ES. Ionising radiation and cancer risks: what have we learned from epidemiology?. Int J Radiat Biol. 2009;85(6):467-482. doi:10.1080/09553000902883836”